# EquiformerV2: Improved Equivariant Transformer for Scaling to Higher-Degree Representations

## Abstract

Equivariant Transformers such as Equiformer have demonstrated the efficacy of applying Transformers to the domain of 3D atomistic systems. However, they are still limited to small degrees of equivariant representations due to their computational complexity. In this paper, we investigate whether these architectures can scale well to higher degrees. Starting from Equiformer, we first replace $SO(3)$ convolutions with eSCN convolutions to efficiently incorporate higher-degree tensors. Then, to better leverage the power of higher degrees, we propose three architectural improvements – attention re-normalization, separable $S^2$ activation and separable layer normalization. Putting this all together, we propose EquiformerV2, which outperforms previous state-of-the-art methods on the large-scale OC20 dataset by up to $15\%$ on forces, $5\%$ on energies, offers better speed-accuracy trade-offs, and $2\times$ reduction in DFT calculations needed for computing adsorption energies.

## 1 Introduction

In recent years, machine learning (ML) models have shown promising results in accelerating and scaling high-accuracy but compute-intensive quantum mechanical calculations by effectively accounting for key features of atomic systems, such as the discrete nature of atoms, and Euclidean and permutation symmetries [1–10]. By bringing down computational costs from hours or days to fractions of seconds, these methods enable new insights in many applications such as molecular simulations, material design and drug discovery. A promising class of ML models that have enabled this progress is equivariant graph neural networks (GNNs) [5, 11–18].

Equivariant GNNs treat 3D atomistic systems as graphs, and incorporate inductive biases such that their internal representations and predictions are equivariant to 3D translations, rotations and optionally inversions. Specifically, they build up equivariant features of each node as vector spaces of irreducible representations (or irreps) and have interactions or message passing between nodes based on equivariant operations such as tensor products. Recent works on equivariant Transformers, specifically Equiformer [17], have shown the efficacy of applying Transformers [19, 20], which have previously enjoyed widespread success in computer vision [21–23], language [24, 25], and graphs [26–29], to this domain of 3D atomistic systems.

A bottleneck in scaling Equiformer as well as other equivariant GNNs is the computational complexity of tensor products, especially when we increase the maximum degree of irreps $L_{max}$. This limits these models to use small values of $L_{max}$ (e.g., $L_{max} \leqslant 3$), which consequently limits their performance. Higher degrees can better capture angular resolution and directional information, which is critical to accurate prediction of atomic energies and forces. To this end, eSCN [18] recently proposes efficient convolutions to reduce $SO(3)$ tensor products to $SO(2)$ linear operations, bringing down the computational cost from $\mathcal{O}(L_{max}^6)$ to $\mathcal{O}(L_{max}^3)$ and enabling scaling to larger values of $L_{max}$ (e.g., $L_{max}$ up to 8). However, except using efficient convolutions for higher $L_{max}$, eSCN still follows SEGNN [15]-like message passing network design, and Equiformer has been shown to improve upon SEGNN. Additionally, this ability to use higher $L_{max}$ challenges whether the previous design of equivariant Transformers can scale well to higher-degree representations.

Submitted to 37th Conference on Neural Information Processing Systems (NeurIPS 2023). Do not distribute.

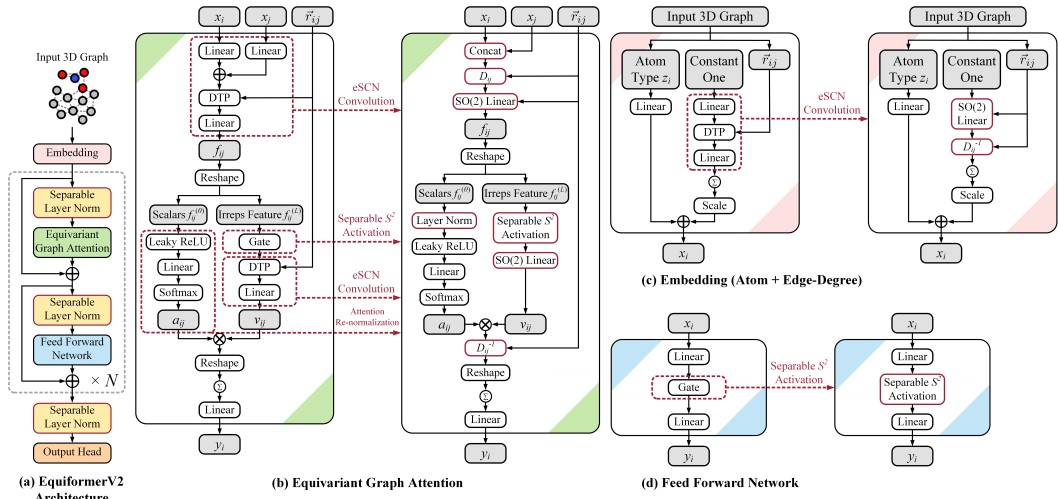

Figure 1: Overview of EquiformerV2. We highlight the differences from Equiformer [17] in red. For (b), (c), and (d), the left figure is the original module in Equiformer, and the right figure is the revised module in EquiformerV2. Input 3D graphs are embedded with atom and edge-degree embeddings and processed with Transformer blocks, which consist of equivariant graph attention and feed forward networks. "⊗" denotes multiplication, "⊕" denotes addition, and $\sum$ within a circle denotes summation over all neighbors. "DTP" denotes depth-wise tensor products used in Equiformer. Gray cells indicate intermediate irreps features.

In this paper, we are interested in adapting eSCN convolutions for higher-degree representations to equivariant Transformers. We start with Equiformer [17] and replace $SO(3)$ convolutions with eSCN convolutions. We find that naively incorporating eSCN convolutions does not result in better performance than the original eSCN model. Therefore, to better leverage the power of higher degrees, we propose three architectural improvements – attention re-normalization, separable $S^2$ activation and separable layer normalization. Putting this all together, we propose EquiformerV2, which is developed on the large and diverse OC20 dataset [30]. Experiments on OC20 show that EquiformerV2 outperforms previous state-of-the-art methods with improvements of up to $15\%$ on forces and $5\%$ on energies, and offers better speed-accuracy trade-offs compared to existing invariant and equivariant GNNs. Additionally, when used in the AdsorbML algorithm [10] for performing adsorption energy calculations, EquiformerV2 achieves the highest success rate and $2\times$ reduction in DFT calculations to achieve comparable adsorption energy accuracies as previous methods.

## 2   Related Works

***SE(3)/E(3)*-Equivariant GNNs.**   Equivariant neural networks [5, 7, 11–18, 31–38] use equivariant irreps features built from vector spaces of irreducible representations (irreps) to achieve equivariance to 3D rotation [11–13]. They operate on irreps features with equivariant operations like tensor products. Previous works differ in equivariant operations used in their networks and how they combine those operations. TFN [11] and NequIP [5] use equivariant graph convolution with linear messages built from tensor products, with the latter utilizing extra equivariant gate activation [12]. SEGNN [15] introduces non-linearity to messages passing [1,39] with equivariant gate activation, and the non-linear messages improve upon linear messages. SE(3)-Transformer [14] adopts equivariant dot product attention [19] with linear messages. Equiformer [17] improves upon previously mentioned equivariant GNNs by combining MLP attention and non-linear messages. Equiformer additionally introduces equivariant layer normalization and regularizations like dropout [40] and stochastic depth [41]. However, the networks mentioned above rely on compute-intensive $SO(3)$ tensor products to mix the information of vectors of different degrees during message passing, and therefore they are limited to small values for maximum degrees $L_{max}$ of equivariant representations. SCN [42] proposes rotating irreps features based on relative position vectors and identifies a subset of spherical harmonics coefficients, on which they can apply unconstrained functions. They further propose relaxing the requirement for strict equivariance and apply typical functions to rotated features during message passing, which trades strict equivariance for computational efficiency and enables using higher values of $L_{max}$. eSCN [18] further improves upon SCN by replacing typical functions with $SO(2)$ linear layers for rotated features and imposing strict equivariance during message passing.

However, except using more efficient operations for higher $L_{max}$, SCN and eSCN mainly adopt the same network design as SEGNN, which is less performant than Equiformer. In this work, we propose EquiformerV2 which includes all the benefits of the above networks by incorporating eSCN convolutions into Equiformer and adopts three additional architectural improvements.

**Invariant GNNs.** Prior works [4, 8, 43–51] extract invariant information from 3D atomistic graphs and operate on the resulting graphs augmented with invariant features. Their differences lie in leveraging different geometric features such as distances, bond angles (3 atom features) or dihedral angles (4 atom features). SchNet [43] models interaction between atoms with only relative distances. DimeNet series [4, 46] use triplet representations of atoms to incorporate bond angles. SphereNet [48] and GemNet [50, 51] further include dihedral angles by considering quadruplet representations. However, the memory complexity of triplet and quadruplet representations of atoms do not scale well with the number of atoms, and this requires additional modifications like interaction hierarchy used by GemNet-OC [51] for large datasets like OC20 [30]. Additionally, for the task of predicting DFT calculations of energies and forces on the large-scale OC20 dataset, invariant GNNs have been surpassed by equivariant GNNs recently.

## 3 Background

### 3.1 $SE(3)/E(3)$-Equivariant Neural Networks

We discuss the relevant background of $SE(3)/E(3)$-equivariant neural networks here. Please refer to Sec. A in appendix for more details of equivariance and group theory.

Including equivariance in neural networks can serve as a strong prior knowledge, which can therefore improve data efficiency and generalization. Equivariant neural networks use equivariant irreps features built from vector spaces of irreducible representations (irreps) to achieve equivariance to 3D rotation. Specifically, the vector spaces are $(2L + 1)$-dimensional, where degree $L$ is a non-negative integer. $L$ can be intuitively interpreted as the angular frequency of the vectors, i.e., how fast the vectors rotate with respect to a rotation of the coordinate system. Higher $L$ is critical to tasks sensitive to angular information like predicting forces [5, 18, 42]. Vectors of degree $L$ are referred to as type-$L$ vectors, and they are rotated with Wigner-D matrices $D^{(L)}$ when rotating coordinate systems. Euclidean vectors $\vec{r}$ in $\mathbb{R}^3$ can be projected into type-$L$ vectors by using spherical harmonics $Y^{(L)}(\frac{\vec{r}}{||\vec{r}||})$. We use order $m$ to index the elements of type-$L$ vectors, where $-L \leqslant m \leqslant L$. We concatenate multiple type-$L$ vectors to form an equivariant irreps feature $f$. Concretely, $f$ has $C_L$ type-$L$ vectors, where $0 \leqslant L \leqslant L_{max}$ and $C_L$ is the number of channels for type-$L$ vectors. In this work, we mainly consider $C_L = C$, and the size of $f$ is $(L_{max} + 1)^2 \times C$. We index $f$ by channel $i$, degree $L$, and order $m$ and denote as $f_{m,i}^{(L)}$.

Equivariant GNNs update irreps features by passing messages of transformed irreps features between nodes. To interact different type-$L$ vectors during message passing, we use tensor products, which generalize multiplication to equivariant irreps features. Denoted as $\otimes_{L_1, L_2}^{L_3}$, the tensor product uses Clebsch-Gordan coefficients to combine type-$L_1$ vector $f^{(L_1)}$ and type-$L_2$ vector $g^{(L_2)}$ and produces type-$L_3$ vector $h^{(L_3)}$:

$$h_{m_3}^{(L_3)} = (f^{(L_1)} \otimes_{L_1, L_2}^{L_3} g^{(L_2)})_{m_3} = \sum_{m_1=-L_1}^{L_1} \sum_{m_2=-L_2}^{L_2} C_{(L_1,m_1)(L_2,m_2)}^{(L_3,m_3)} f_{m_1}^{(L_1)} g_{m_2}^{(L_2)} \tag{1}$$

where $m_1$ denotes order and refers to the $m_1$-th element of $f^{(L_1)}$. Clebsch-Gordan coefficients $C_{(L_1,m_1)(L_2,m_2)}^{(L_3,m_3)}$ are non-zero only when $|L_1 - L_2| \leqslant L_3 \leqslant |L_1 + L_2|$ and thus restrict output vectors to be of certain degrees. We typically discard vectors with $L > L_{max}$, where $L_{max}$ is a hyper-parameter, to prevent vectors of increasingly higher dimensions. In many works, message passing is implemented as equivariant convolutions, which perform tensor products between input irreps features $x^{(L_1)}$ and spherical harmonics of relative position vectors $Y^{(L_2)}(\frac{\vec{r}}{||\vec{r}||})$.

### 3.2 Equiformer

Equiformer [17] is an $SE(3)/E(3)$-equivariant GNN that combines the inductive biases of equivariance with the strength of Transformers [19, 22]. First, Equiformer replaces scalar node features with equivariant irreps features to incorporate equivariance. Next, it performs equivariant operations on these irreps features and equivariant graph attention for message passing. These operations include

tensor products and equivariant linear operations, equivariant layer normalization [52] and gate activation [12, 34]. For stronger expressivity in the attention compared to typical Transformers, Equiformer uses non-linear functions for both attention weights and message passing. Additionally, Equiformer incorporates regularization techniques common in Transformers applied to other domains, e.g., dropout [40] to attention weights [53] and stochastic depth [54] to the outputs of equivariant graph attention and feed forward networks. Please refer to the Equiformer paper [17] for more details.

### 3.3 eSCN Convolution

While tensor products are necessary to interact vectors of different degrees, they are compute-intensive. To reduce the complexity, eSCN convolutions [18] are proposed to use $SO(2)$ linear operations for efficient tensor products. We provide an outline and intuition for their method here, and please refer to Sec. A and their work [18] for mathematical details.

A traditional $SO(3)$ convolution interacts input irreps features $x_{m_i}^{(L_i)}$ and spherical harmonic projec-
tions of relative positions $Y_{m_f}^{(L_f)}(\vec{r}_{ij})$ with an $SO(3)$ tensor product with Clebsch-Gordan coefficients
$C_{(L_i,m_i),(L_f,m_f)}^{(L_o,m_o)}$. The projection $Y_{m_f}^{(L_f)}(\vec{r}_{ij})$ becomes sparse if we rotate the relative position vector
$\vec{r}_{ij}$ with a rotation matrix $D_{ij}$ to align with the direction of $L = 0$ and $m = 0$, which corresponds to
the z axis traditionally but the y axis in the conventions of e3nn [55]. Concretely, given $D_{ij}\vec{r}_{ij}$ aligned
with the y axis, $Y_{m_f}^{(L_f)}(D_{ij}\vec{r}_{ij}) \neq 0$ only for $m_f = 0$. If we consider only $m_f = 0$, $C_{(L_i,m_i),(L_f,m_f)}^{(L_o,m_o)}$
can be simplified, and $C_{(L_i,m_i),(L_f,0)}^{(L_o,m_o)} \neq 0$ only when $m_i = \pm m_o$. Therefore, the original expression
depending on $m_i$, $m_f$, and $m_o$ is now reduced to only depend on $m_o$. This means we are no longer
mixing all integer values of $m_i$ and $m_f$, and outputs of order $m_o$ are linear combinations of inputs
of order $\pm m_o$. eSCN convolutions go one step further and replace the remaining non-trivial paths
of the $SO(3)$ tensor product with an $SO(2)$ linear operation to allow for additional parameters of
interaction between $\pm m_o$ without breaking equivariance. To summarize, eSCN convolutions achieve
efficient equivariant convolutions by first rotating irreps features based on relative position vectors
and then performing $SO(2)$ linear operations on the rotated features. The key idea is that the rotation
sparsifies tensor products and simplifies the computation.

## 4 EquiformerV2

Starting from Equiformer [17], we first use eSCN convolutions to scale to higher-degree representations (Sec. 4.1). Then, we propose three architectural improvements, which yield further performance gain when using higher degrees: attention re-normalization (Sec. 4.2), separable $S^2$ activation (Sec. 4.3) and separable layer normalization (Sec. 4.4). Figure 1 illustrates the overall architecture of EquiformerV2 and the differences from Equiformer.

### 4.1 Incorporating eSCN Convolutions for Efficient Tensor Products and Higher Degrees

The computational complexity of $SO(3)$ tensor products used in traditional $SO(3)$ convolutions during equivariant message passing scale unfavorably with $L_{max}$. Because of this, it is impractical for Equiformer to use beyond $L_{max} = 1$ for large-scale datasets like OC20 [30] and beyond $L_{max} = 3$ for small-scale datasets like MD17 [56–58]. Since higher $L_{max}$ can better capture angular information and are correlated with model expressivity [5], low values of $L_{max}$ can lead to limited performance on certain tasks such as predicting forces. Therefore, we replace original tensor products with eSCN convolutions [18] for efficient tensor products, enabling Equiformer to scale up $L_{max}$ to 6 or 8 on the large-scale OC20 dataset.

Equiformer uses equivariant graph attention for message passing. The attention consists of depth-wise tensor products, which mix information across different degrees, and linear layers, which mix information between channels of the same degree. Since eSCN convolutions mix information across both degrees and channels, we replace the $SO(3)$ convolution, which involves one depth-wise tensor product layer and one linear layer, with a single eSCN convolutional layer, which consists of a rotation matrix $D_{ij}$ and an $SO(2)$ linear layer as shown in Figure 1b.

### 4.2 Attention Re-normalization

Equivariant graph attention in Equiformer uses tensor products to project node embeddings $x_i$ and $x_j$,
which contain vectors of different degrees, to scalar features $f_{ij}^{(0)}$ and applies non-linear functions to
$f_{ij}^{(0)}$ for attention weights $a_{ij}$. The node embeddings $x_i$ and $x_j$ are obtained by applying equivariant

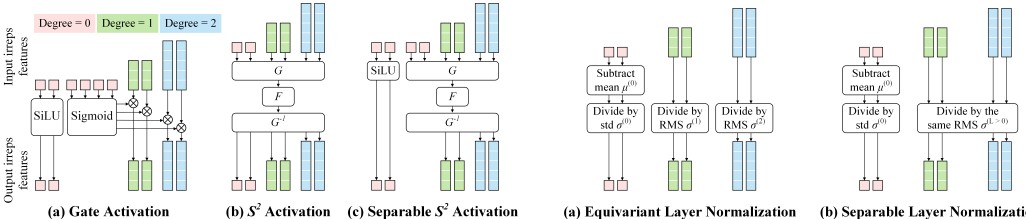

Figure 2: Illustration of different activation functions. $G$ denotes conversion from vectors to point samples on a sphere, $F$ can typically be a SiLU activation or MLPs, and $G^{-1}$ is the inverse of $G$.

Figure 3: Illustration of how statistics are calculated in different normalizations. "std" denotes standard deviation, and "RMS" denotes root mean square.

layer normalization [17] to previous outputs. We note that vectors of different degrees in $x_i$ and $x_j$ are normalized independently, and therefore when they are projected to the same degree, the resulting $f_{ij}^{(0)}$ can be less well-normalized. To address the issue, we propose attention re-normalization and introduce one additional layer normalization (LN) [52] before non-linear functions. Specifically, given $f_{ij}^{(0)}$, we first apply LN and then use one leaky ReLU layer and one linear layer to calculate $z_{ij} = a^\top \text{LeakyReLU}(\text{LN}(f_{ij}^{(0)}))$ and $a_{ij} = \text{softmax}_j(z_{ij}) = \frac{\exp(z_{ij})}{\sum_{k \in \mathcal{N}(i)} \exp(z_{ik})}$, where $a$ is a learnable vector of the same dimension as $f_{ij}^{(0)}$.

### 4.3 Separable $S^2$ Activation

The gate activation [12] used by Equiformer applies sigmoid activation to scalar features to obtain non-linear weights and then multiply irreps features of degree $> 0$ with non-linear weights to add non-linearity to equivariant features. The activation, however, only accounts for the interaction from vectors of degree $0$ to those of degree $> 0$ and could be sub-optimal when we scale up $L_{max}$.

To better mix the information across degrees, SCN [42] and eSCN [18] propose to use $S^2$ activation [59]. The activation first converts vectors of all degrees to point samples on a sphere for each channel, applies unconstrained functions $F$ to those samples, and finally convert them back to vectors. Specifically, given an input irreps feature $x \in \mathbb{R}^{(L_{max}+1)^2 \times C}$, the output is $y = G^{-1}(F(G(x)))$, where $G$ denotes the conversion from vectors to point samples on a sphere, $F$ can be typical SiLU activation [60, 61] or typical MLPs, and $G^{-1}$ is the inverse of $G$.

While $S^2$ activation can better mix vectors of different degrees, we find that directly replacing the gate activation with $S^2$ activation results in training instability (row 3 in Table 1a). To address the issue, we propose separable $S^2$ activation, which separates activation for vectors of degree $0$ and those of degree $> 0$. Similar to gate activation, we have more channels for vectors of degree $0$. As shown in Figure 2c, we apply a SiLU activation to the first part of vectors of degree $0$, and the second part of vectors of degree $0$ are used for $S^2$ activation along with vectors of higher degrees. After $S^2$ activation, we concatenate the first part of vectors of degree $0$ with vectors of degrees $> 0$ as the final output and ignore the second part of vectors of degree $0$. Additionally, we also use separable $S^2$ activation in point-wise feed forward networks (FFNs). Figure 2 illustrates the differences between gate activation, $S^2$ activation and separable $S^2$ activation.

### 4.4 Separable Layer Normalization

As mentioned in Sec. 4.2, equivariant layer normalization used by Equiformer normalizes vectors of different degrees independently, and when those vectors are projected to the same degree, the projected vectors can be less well-normalized. Therefore, instead of performing normalization to each degree independently, we propose separable layer normalization (SLN), which separates normalization for vectors of degree $0$ and those of degrees $> 0$. Mathematically, let $x \in \mathbb{R}^{(L_{max}+1)^2 \times C}$ denote an input irreps feature of maximum degree $L_{max}$ and $C$ channels, and $x_{m,i}^{(L)}$ denote the $L$-th degree, $m$-th order and $i$-th channel of $x$. SLN calculates the output $y$ as follows. For $L = 0$, $y^{(0)} = \gamma^{(0)} \circ \left( \frac{x^{(0)} - \mu^{(0)}}{\sigma^{(0)}} \right) + \beta^{(0)}$, where $\mu^{(0)} = \frac{1}{C} \sum_{i=1}^{C} x_{0,i}^{(0)}$ and $\sigma^{(0)} = \sqrt{\frac{1}{C} \sum_{i=1}^{C} (x_{0,i}^{(0)} - \mu^{(0)})^2}$. For $L > 0$, $y^{(L)} = \gamma^{(L)} \circ \left( \frac{x^{(L)}}{\sigma^{(L>0)}} \right)$, where $\sigma^{(L>0)} = \sqrt{\frac{1}{L_{max}} \sum_{L=1}^{L_{max}} \left( \sigma^{(L)} \right)^2}$ and $\sigma^{(L)} = \sqrt{\frac{1}{C} \sum_{i=1}^{C} \frac{1}{2L+1} \sum_{m=-L}^{L} \left( x_{m,i}^{(L)} \right)^2}$.

$\gamma^{(0)}, \gamma^{(L)}, \beta^{(0)} \in \mathbb{R}^C$ are learnable parameters, $\mu^{(0)}$ and $\sigma^{(0)}$ are mean and standard deviation of vectors of degree 0, $\sigma^{(L)}$ and $\sigma^{(L>0)}$ are root mean square values (RMS), and $\circ$ denotes element-wise product. The computation of $y^{(0)}$ corresponds to typical layer normalization. We note that the difference between equivariant layer normalization and SLN lies only in $y^{(L)}$ with $L > 0$ and that equivariant layer normalization divides $x^{(L)}$ by $\sigma^{(L)}$, which is calculated independently for each degree $L$, instead of $\sigma^{(L>0)}$, which considers all degrees $L > 0$. Figure 3 compares how $\mu^{(0)}$, $\sigma^{(0)}$, $\sigma^{(L)}$ and $\sigma^{(L>0)}$ are calculated in equivariant layer normalization and SLN.

### 4.5  Overall Architecture

Here, we discuss all the other modules in EquiformerV2 and focus on the differences from Equiformer.

**Equivariant Graph Attention.**   Figure 1b illustrates equivariant graph attention after the above modifications. As described in Sec. 4.1, given node embeddings $x_i$ and $x_j$, we first concatenate them along the channel dimension and then rotate them with rotation matrices $D_{ij}$ based on their relative positions or edge directions $\vec{r}_{ij}$. The rotation enables reducing $SO(3)$ tensor products to $SO(2)$ linear operations, and we replace depth-wise tensor products and linear layers between $x_i$, $x_j$ and $f_{ij}$ with a single $SO(2)$ linear layer. To consider the information of relative distances $||\vec{r}_{ij}||$, in the same way as eSCN [18], we transform $||\vec{r}_{ij}||$ with a radial function to obtain distance embeddings and then multiply distance embeddings with concatenated node embeddings before the first $SO(2)$ linear layer. We split the outputs $f_{ij}$ of the first $SO(2)$ linear layer into two parts. The first part is scalar features $f_{ij}^{(0)}$, which only contains vectors of degree 0, and the second part is irreps features $f_{ij}^{(L)}$ and includes vectors of all degrees up to $L_{max}$. As mentioned in Sec. 4.2, we first apply an additional LN to $f_{ij}^{(0)}$ and then follow the design of Equiformer by applying one leaky ReLU layer, one linear layer and a final softmax layer to obtain attention weights $a_{ij}$. As for value $v_{ij}$, we replace the gate activation with separable $S^2$ activation with $F$ being a single SiLU activation and then apply the second $SO(2)$ linear layer. While in Equiformer, the message $m_{ij}$ sent from node $j$ to node $i$ is $m_{ij} = a_{ij} \times v_{ij}$, here we need to rotate $a_{ij} \times v_{ij}$ back to original coordinate frames and the message $m_{ij}$ becomes $D_{ij}^{-1}(a_{ij} \times v_{ij})$. Finally, we can perform $h$ parallel equivariant graph attention functions given $f_{ij}$. The $h$ different outputs are concatenated and projected with a linear layer to become the final output $y_i$. Parallelizing attention functions and concatenating can be implemented with "Reshape".

**Feed Forward Network.**   As illustrated in Figure 1d, we replace the gate activation with separable $S^2$ activation. The function $F$ consists of a two-layer MLP, with each linear layer followed by SiLU, and a final linear layer.

**Embedding.**   This module consists of atom embedding and edge-degree embedding. The former is the same as that in Equiformer. For the latter, as depicted in the right branch in Figure 1c, we replace original linear layers and depth-wise tensor products with a single $SO(2)$ linear layer followed by a rotation matrix $D_{ij}^{-1}$. Similar to equivariant graph attention, we consider the information of relative distances by multiplying the outputs of the $SO(2)$ linear layer with distance embeddings.

**Radial Basis and Radial Function.**   We represent relative distances $||\vec{r}_{ij}||$ with a finite radial basis like Gaussian radial basis functions [43] to capture their subtle changes. We transform radial basis with a learnable radial function to generate distance embeddings. The function consists of a two-layer MLP, with each linear layer followed by LN and SiLU, and a final linear layer.

**Output Head.**   To predict scalar quantities like energy, we use one feed forward network to transform irreps features on each node into a scalar and then perform sum aggregation over all nodes. As for predicting forces acting on each node, we use a block of equivariant graph attention and treat the output of degree 1 as our predictions.

## 5  OC20 Experiments

Our experiments focus on the large and diverse OC20 dataset [30] (Creative Commons Attribution 4.0 License), which consists of 1.2M DFT relaxations for training and evaluation, computed with the revised Perdew-Burke-Ernzerhof (RPBE) functional [62]. Each structure in OC20 has an adsorbate molecule placed on a catalyst surface, and the core task is Structure-to-Energy-Forces (S2EF), which is to predict the energy of the structure and per-atom forces. Models trained for the S2EF task are evaluated on energy and force mean absolute error (MAE). These models can in turn be used for performing structure relaxations by using the model's force predictions to iteratively update the atomic positions until a relaxed structure corresponding to a local energy minimum is found. These

| | Attention Re-normalization | Activation | Normalization | Epochs | forces | energy |
|---|---|---|---|---|---|---|
| 1 | ✗ | Gate | LN | 12 | 21.85 | 286 |
| 2 | ✓ | Gate | LN | 12 | 21.86 | 279 |
| 3 | ✓ | $S^2$ | LN | 12 | didn't converge | |
| 4 | ✓ | Sep. $S^2$ | LN | 12 | 20.77 | 285 |
| 5 | ✓ | Sep. $S^2$ | SLN | 12 | 20.46 | 285 |
| 6 | ✓ | Sep. $S^2$ | LN | 20 | 20.02 | 276 |
| 7 | ✓ | Sep. $S^2$ | SLN | 20 | 19.72 | 278 |
| 8 | eSCN baseline | | | 12 | 21.3 | 294 |

(a) Architectural improvements. Attention re-normalization improves energies, and separable $S^2$ activation ("Sep. $S^2$") and separable layer normalization ("SLN") improve forces.

| | | eSCN | | EquiformerV2 | |
|---|---|---|---|---|---|
| $L_{max}$ | Epochs | forces | energy | forces | energy |
| 6 | 12 | 21.3 | 294 | 20.46 | 285 |
| 6 | 20 | 20.6 | 290 | 19.78 | 280 |
| 6 | 30 | 20.1 | 285 | 19.42 | 278 |
| 8 | 12 | 21.3 | 296 | 20.46 | 279 |
| 8 | 20 | - | - | 19.95 | 273 |

(b) Training epochs. Training for more epochs consistently leads to better results.

| | eSCN | | EquiformerV2 | |
|---|---|---|---|---|
| $L_{max}$ | forces | energy | forces | energy |
| 4 | 22.2 | 291 | 21.37 | 284 |
| 6 | 21.3 | 294 | 20.46 | 285 |
| 8 | 21.3 | 296 | 20.46 | 279 |

(c) Degrees $L_{max}$. Higher degrees are consistently helpful.

| | eSCN | | EquiformerV2 | |
|---|---|---|---|---|
| $M_{max}$ | forces | energy | forces | energy |
| 2 | 21.3 | 294 | 20.46 | 285 |
| 3 | 21.2 | 295 | 20.24 | 284 |
| 4 | 21.2 | 298 | 20.24 | 282 |
| 6 | - | - | 20.26 | 278 |

(d) Orders $M_{max}$. Higher orders mainly improve energy predictions.

| | eSCN | | EquiformerV2 | |
|---|---|---|---|---|
| Layers | forces | energy | forces | energy |
| 8 | 22.4 | 306 | 21.18 | 293 |
| 12 | 21.3 | 294 | 20.46 | 285 |
| 16 | 20.5 | 283 | 20.11 | 282 |

(e) Number of blocks. Adding more Transformer blocks can help both force and energy predictions.

Table 1: Ablation results with EquiformerV2. We report mean absolute errors for forces in meV/Å and energy in meV, and lower is better. All models are trained on the 2M subset of OC20 [30], and errors are averaged over the four validation splits of OC20. The base model setting is marked in gray.

relaxed structure and energy predictions are evaluated on the Initial Structure to Relaxed Structure (IS2RS) and Initial Structure to Relaxed Energy (IS2RE) tasks. The "All" split of OC20 contains 134M training structures spanning 56 elements, and "MD" split consists of 38M structures. We first conduct ablation studies on EquiformerV2 trained on the smaller S2EF-2M subset (Sec. 5.1). Then, we report the results of training on S2EF-All and S2EF-All+MD splits (Sec. 5.2). Additionally, we investigate the performance of EquiformerV2 when used in the AdsorbML algorithm [10] (Sec. 5.3). Please refer to Sec. B and C for details of models and training.

## 5.1 Ablation Studies

**Architectural Improvements**. In Table 1a, we ablate the three proposed architectural changes – attention re-normalization, separable $S^2$ activation and separable layer normalization. First, with attention re-normalization (row 1 and 2), energy errors improve by 2.4%, while force errors are about the same. Next, we replace the gate activation with $S^2$ activation used in SCN [42] and eSCN [18], but that does not converge (row 3). Instead, using the proposed separable $S^2$ activation (row 4), where we have separate paths for invariant and equivariant features, converges to 5% better forces albeit hurting energies. Similarly, replacing equivariant layer normalization with separable layer normalization (row 5) further improves forces by 1.5%. Finally, these modifications enable training for longer without overfitting (row 7), further improving forces by 3.6% and recovering energies to similar accuracies as Index 2. Overall, our modifications improve forces by 10% and energies by 3%. Note that simply incorporating eSCN convolutions into Equiformer (row 1) and using higher degrees does not result in improving over the original eSCN baseline (row 8), and that the proposed architectural changes are necessary.

**Scaling of Parameters**. In Tables 1c, 1d, 1e, we systematically vary the maximum degree $L_{max}$, the maximum order $M_{max}$, and the number of Transformer blocks and compare with equivalent eSCN variants. There are several key takeaways. First, across all experiments, EquiformerV2 performs better than its eSCN counterparts. Second, while one might intuitively expect higher resolution features and larger models to perform better, this is only true for EquiformerV2, not eSCN. For example, increasing $L_{max}$ from 6 to 8 or $M_{max}$ from 3 to 4 degrades the performance of eSCN on energy predictions but helps that of EquiformerV2. In Table 1b, we show that longer training regimes are crucial. Increasing the training epochs from 12 to 30 with $L_{max} = 6$ improves force and energy predictions by 5% and 2.5%, respectively.

**Comparison of Speed-Accuracy Trade-offs**. To be practically useful for atomistic simulations and material screening, models should offer flexibility in speed-accuracy tradeoffs. We compare these trade-offs for EquiformerV2 with prior works in Figure 4a. Here, the speed is reported as the number of structures processed per GPU-second during inference and measured on V100 GPUs.

| Training set | Model | Throughput Samples / GPU sec. ↑ | S2EF validation Energy MAE (meV) ↓ | Force MAE (meV/Å) ↓ | S2EF test Energy MAE (meV) ↓ | Force MAE (meV/Å) ↓ | IS2RS test AFbT (%) ↑ | ADwT (%) ↑ | IS2RE test Energy MAE (meV) ↓ |
|---|---|---|---|---|---|---|---|---|---|
| OC20 All | CGCNN [44] | - | 590 | 74.0 | 608 | 73.3 | - | - | - |
| | SchNet [43] | - | 549 | 56.8 | 540 | 54.7 | - | 14.4 | 764 |
| | ForceNet-large [63] | 15.3 | - | 33.5 | - | 32.0 | 12.7 | 49.6 | - |
| | DimeNet++-L-F+E [4] | 4.6 | 515 | 32.8 | 480 | 31.3 | 21.7 | 51.7 | 559 |
| | SpinConv [49] | 6.0 | 371 | 41.2 | 336 | 29.7 | 16.7 | 53.6 | 437 |
| | GemNet-dT [50] | 25.8 | 315 | 27.2 | 292 | 24.2 | 27.6 | 58.7 | 400 |
| | GemNet-XL [8] | 1.5 | - | - | 270 | 20.5 | 30.8 | 62.7 | 371 |
| | GemNet-OC [51] | 18.3 | 244 | 21.7 | 233 | 20.7 | 35.3 | 60.3 | 355 |
| | SCN L=8 K=20 [42] | - | - | - | 244 | 17.7 | 40.3 | 67.1 | 330 |
| | eSCN L=6 K=20 [18] | 2.9 | - | - | 242 | 17.1 | 48.5 | 65.7 | 341 |
| | EquiformerV2 ($\lambda_E = 2$) | 1.8 | **236** | **15.7** | 229 | 14.8 | **53.0** | **69.0** | **316** |
| OC20 All+MD | GemNet-OC-L-E [51] | 7.5 | 239 | 22.1 | 230 | 21.0 | - | - | - |
| | GemNet-OC-L-F [51] | 3.2 | 252 | 20.0 | 241 | 19.0 | 40.6 | 60.4 | - |
| | GemNet-OC-L-F+E [51] | - | - | - | - | - | - | - | 348 |
| | SCN L=6 K=16 (4-tap 2-band) [42] | - | - | - | 228 | 17.8 | 43.3 | 64.9 | 328 |
| | SCN L=8 K=20 [42] | - | - | - | 237 | 17.2 | 43.6 | 67.5 | 321 |
| | eSCN L=6 K=20 [18] | 2.9 | 243 | 17.1 | 236 | 16.2 | 50.3 | 66.7 | 327 |
| | EquiformerV2 ($\lambda_E = 2$) | 1.8 | 230 | **14.6** | 227 | **13.8** | **55.4** | **69.8** | 311 |
| | EquiformerV2 ($\lambda_E = 4$) | 1.8 | **227** | 15.0 | **219** | 14.2 | 54.4 | 69.4 | **309** |

Table 2: OC20 results on S2EF validation and test splits, and IS2RS and IS2RE test splits when trained on OC20 S2EF-All or S2EF-All+MD splits. Throughput is reported as the number of structures processed per GPU-second during training and measured on V100 GPUs. $\lambda_E$ is the coefficient of the energy loss.

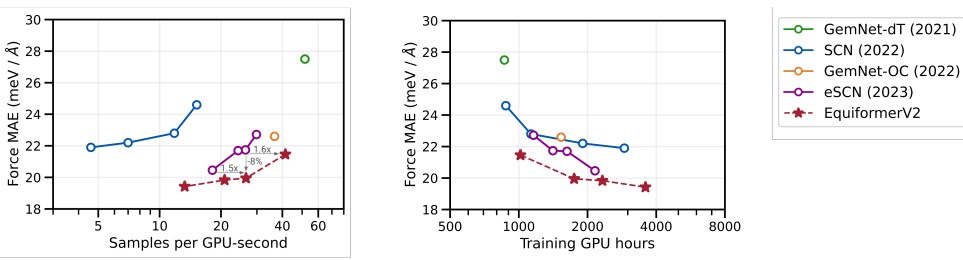

(a) Trade-offs between inference speed and validation force MAE.

(b) Trade-offs between training cost and validation force MAE.

Figure 4: EquiformerV2 offers better accuracy trade-offs both in terms of inference speed as well as training cost compared to prior works. All models in this analysis are trained on the S2EF-2M split.

For the same force MAE as eSCN, EquiformerV2 is up to $1.6\times$ faster, and for the same speed as eSCN, EquiformerV2 is up to $8\%$ more accurate. Compared to GemNet-OC [51] at the same speed, EquiformerV2 is $5\%$ more accurate. Comparing to the closest available EquifomerV2 point, GemNet-dT [50] is $1.25\times$ faster but $30\%$ worse. Overall, EquiformerV2 clearly offers a better trade-off between speed and accuracy. In similar spirit, we also study the training cost of EquiformerV2 compared to prior works in Figure 4b, and find that it is substantially more training efficient.

## 5.2 Main Results

Table 2 reports results on the test splits for all the three tasks of OC20, averaged across the in-distribution, out-of-distribution adsorbates, out-of-distribution catalysts, and out-of-distribution both subsplits. Models are trained on either OC20 S2EF-All and S2EF-All+MD splits. All test results are computed via the EvalAI evaluation server[1]. EquiformerV2 outperforms all previous models across all tasks, improving by $4\%$ on S2EF energy MAE, by $15\%$ on S2EF force MAE, by $5\%$ absolute on IS2RS Average Forces below Threshold (AFbT), and by $4\%$ on IS2RE energy MAE. In particular, the improvements in force predictions are significant. Going from SCN [42] to eSCN [18], S2EF test force MAE improves from 17.2 meV/Å to 16.2 meV/Å , largely due to replacing approximate equivariance in SCN with strict equivariance in eSCN during message passing and scaling to higher degrees. Similarly, by scaling up the degrees of representations in Equiformer [17], EquiformerV2 further improves force MAE to 13.8 meV/Å, more than doubling the gain of going from SCN to eSCN. These better force predictions also translate to higher IS2RS test AFbT, which is computed via DFT single-point calculations to check if the DFT forces on the predicted relaxed structures are close to zero. A $5\%$ improvement on AFbT is a strong step towards replacing DFT with ML.

## 5.3 AdsorbML Results

Lan et al. [10] recently proposed the AdsorbML algorithm, wherein they show that recent state-of-the-art GNNs (e.g. SCN [42]) can achieve more than $1000\times$ speedup over DFT relaxations at computing

---

[1] eval.ai/web/challenges/challenge-page/712

| Model | k = 1 Success | k = 1 Speedup | k = 2 Success | k = 2 Speedup | k = 3 Success | k = 3 Speedup | k = 4 Success | k = 4 Speedup | k = 5 Success | k = 5 Speedup |
|---|---|---|---|---|---|---|---|---|---|---|
| SchNet [43] | 2.77% | 4266.13 | 3.91% | 2155.36 | 4.32% | 1458.77 | 4.73% | 1104.88 | 5.04% | 892.79 |
| DimeNet++ [4] | 5.34% | 4271.23 | 7.61% | 2149.78 | 8.84% | 1435.21 | 10.07% | 1081.96 | 10.79% | 865.20 |
| PaiNN [34] | 27.44% | 4089.77 | 33.61% | 2077.65 | 36.69% | 1395.55 | 38.64% | 1048.63 | 39.57% | 840.44 |
| GemNet-OC [51] | 68.76% | 4185.18 | 77.29% | 2087.11 | 80.78% | 1392.51 | 81.50% | 1046.85 | 82.94% | 840.25 |
| GemNet-OC-MD [51] | 68.76% | 4182.04 | 78.21% | 2092.27 | 81.81% | 1404.11 | 83.25% | 1053.36 | 84.38% | 841.64 |
| GemNet-OC-MD-Large [51] | 73.18% | 4078.76 | 79.65% | 2065.15 | 83.25% | 1381.39 | 85.41% | 1041.50 | 86.02% | 834.46 |
| SCN-MD-Large [42] | 77.80% | 3974.21 | 84.28% | 1989.32 | 86.33% | 1331.43 | 87.36% | 1004.40 | 87.77% | 807.00 |
| EquiformerV2 ($\lambda_E = 4$) | **85.41%** | 4001.71 | **88.90%** | 2012.47 | **90.54%** | 1352.08 | **91.06%** | 1016.31 | **91.57%** | 815.87 |

Table 3: AdsorbML results with EquiformerV2 ($\lambda_E = 4$) trained on S2EF-All+MD from Table 2.

adsorption energies within a $0.1\mathrm{eV}$ margin of DFT results with an $87\%$ success rate. This is done by using OC20-trained models to perform structure relaxations for an average 90 configurations of an adsorbate placed on a catalyst surface, followed by DFT single-point calculations for the top-$k$ structures with lowest predicted relaxed energies, as a proxy for calculating the global energy minimum or adsorption energy. We refer the reader to the AdsorbML paper [10] for more details. We benchmark AdsorbML with EquiformerV2, and Table 3 shows that it improves over SCN by a significant margin, with $8\%$ and $5\%$ absolute improvements at $k = 1$ and $k = 2$, respectively. Moreover, EquiformerV2 at $k = 2$ is more accurate at adsorption energy calculations than all the other models even at $k = 5$, thus requiring at least $2\times$ fewer DFT calculations.

## 6 Conclusion

In this work, we investigate how equivariant Transformers can be scaled up to higher degrees of equivariant representations. We start by replacing $SO(3)$ convolutions in Equiformer with eSCN convolutions, and propose three architectural improvements to better leverage the power of higher degrees – attention re-normalization, separable $S^2$ activation and separable layer normalization. With these modifications, we propose EquiformerV2, which outperforms state-of-the-art methods on the S2EF, IS2RS, and IS2RE tasks on the OC20 dataset, improves speed-accuracy trade-offs, and achieves the best success rate when used in AdsorbML.

**Broader Impacts.** EquiformerV2 achieves more accurate approximation of quantum mechanical calculations and demonstrates one further step toward replacing DFT force fields with machine learned ones. By demonstrating its promising results, we hope to encourage the community to make further progress in applications like material design and drug discovery than to use it for adversarial purposes. Additionally, the method only facilitates identification of molecules or materials of specific properties, and there are substantial hurdles from their large-scale deployment. Finally, we note that the proposed method is general and can be applied to different problems like protein structure prediction [64] as long as inputs can be modeled as 3D graphs.

**Limitations.** Although EquiformerV2 improves upon state-of-the-art methods on the large and diverse OC20 dataset, we acknolwdge that the performance gains brought by scaling to higher degrees and the proposed architectural improvements can depend on tasks and datasets. For example, the increased expressivity may lead to overfitting on smaller datasets like QM9 [65, 66] and MD17 [56–58]. However, the issue can be mitigated by pre-training on large datasets like OC20 [30] and PCQM4Mv2 [67] optionally via denoising [68] and then finetuning on smaller datasets.

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
