# OpenReview forum: "EquiformerV2: Improved Equivariant Transformer for Scaling to Higher-Degree Representations"
_NeurIPS.cc/2023/Conference — Submitted to NeurIPS 2023_

### Official Review · Reviewer_9yvD · 2023-06-23

**Soundness:** 2 fair
**Presentation:** 3 good
**Contribution:** 3 good
**Rating:** 6
**Confidence:** 4

**Summary:**

The authors propose EquiformerV2, which is an improvement over the original Equiformer architecture. The main improvement is using a more efficient parameterization of the tensor products used in Equiformer which is computationally expensive for higher order representations. The more efficient parameterization involves using SO(2) linear layers instead of SO(3) tensor products. Moreover, they have three architectural improvements – adding an extra layer norm during attention, S2 activations instead of gates and "separable" layer normalization which differs from the previous layer normalization in terms of the denominator used. They perform experiments on OC20 to show improvements compared to other models in the literature.

**Strengths:**

Overall, the idea of using computing tensor products more efficiently using SO(2) linear layers makes good sense in its application to Equiformer in order to scale it up via higher order SO(3) representations. Empirical work around improving the architecture by appropriate use of different type of layer norms and activation functions is also valuable. The components feeding into this model aren't always original, but that shouldn't detract from the significance of putting together the full model and showing that it improves performance on a benchmark. The clarity of the writing is usually good but there are places where it can be improved (see below).

**Weaknesses:**

The main weakness is insufficient comparison against the original Equiformer architecture. There are four differences versus the V1 architecture as far as I can tell: (1) the more efficient parameterization of tensor products via the ideas of eSCN, (2) an extra layer norm applied to scalar features in the attention module, (3) separable S2 activation instead of gates and (4) layer normalization with a different way of computing the denominator (called "separable" layer normalization in the paper).

- There is an ablation in Table 1(a) on one of the datasets looking at the effects of changes (2), (3), and (4). It is unclear how much of the difference in performance (specially, on energies) is statistically significant. I understand training each model is expensive, but since the experiments in this table are on the smaller S2EF-2M dataset, isn't it possible to have error bars here?
- Why isn't there a comparison in Table 1(a) of the effect of incorporating change (1) in EquiformerV2? The SO(2) linear layers replacing the tensor product are in principle as expressive as the tensor products but the optimization dynamics can be different since the parameterization of the architecture is different. It would be good to see a side-by-side comparison of Equiformer's V1 and V2 when keeping other architectural hyperparameters fixed (e.g. number of channels, maximum representation order etc.)
- It would be good to include EquiformerV1 performance in the other sub-tables of Table 1 too, in order to see the cummulative effect of the changes to the original architecture.
- Similarly, it would be useful to see EquiformerV1 performance in Table 2 and 3 as well (possibly with a lower $L_\text{max}$ but with other hyperparameters tuned).
- How does the setup for IS2RE in Table 2 differ from EquiformerV1's Table 3? Is one with relaxations after training on S2EF and the other is through direct energy prediction? A side-by-side comparison under the same setup would help a lot here.
- How necessary are the higher orders compared to increasing the number of layers or channels? SO(2) linear layers instead of tensor products will improve the efficiency for lower orders too, so can we make up for the performance of higher orders by instead increasing the number of layers or channels?
- Have you benchmarked EquiformerV2 on QM9 where EquiformerV1 has already been benchmarked? It's a dataset that is different from OC20 in various ways, so it would be useful to see a comparison.

For other suggestions for improvements, see the questions below.

Equiformer(V1): https://arxiv.org/abs/2206.11990

**Questions:**

- The explanation of separable S2 activation was unclear: what does line 186 mean? "The activation first converts vectors of all degrees to point samples on a sphere for each channel, applies unconstrained functions F to those samples, and finally convert them back to vectors."
- Have you done direct IS2RE prediction as done in the EquiformerV1 paper with/without an auxiliary denoising loss?
- Can you justify "higher degrees can better capture angular resolution and directional information"? Perhaps via an example?
- Was stochastic depth used in EquiformerV2? Related work section mentions it was used in EquiformerV1?
- Line 69: "apply typical functions to rotated features during message passing" was unclear to me.
- Line 142: "eSCN convolutions go one step further and replace the remaining non-trivial paths of the SO(3) tensor product with an SO(2) linear operation to allow for additional parameters of interaction between $\pm m_o$ without breaking equivariance." Can you explain this in the context of Appendix A.3?
- Line 175: why would $f^{(0)}_{ij}$ be less well-normalized?
- Line 178: there is a slight overload in notation with $a$ and $a_{ij}$.
- Line 210: why is there no centering in the layer norm formulation? Wouldn't that continue to preserve equivariance as the mean vector belongs to the same representation? There is centering in the 0th degree activations and traditional layer norm.
- Line 210: do you use a small $\epsilon$ in the new separable layer normalization denominator for numerical stability?
- Line 280: "Finally, these modifications enable training for longer without overfitting (row 7)" don't both models improve performance when trained for longer, not just the model in row 7?
- On comparison of speed: are all the models using the same training/inference pipeline such that the comparison is fair? It would be useful to report inference cost in FLOPs or a similar metric as well.




**Limitations:**

Limitations and broader impact are adequately addressed.

---

> ### Author Rebuttal · Authors · 2023-08-10
>
> > 1. [Weakness 1] Statistic significance of numbers in Table 1(a).
>
> We empirically find that training the same models twice with identical settings results in almost the same force MAE with differences less than 0.05 meV/Å.
>
> In comparison, increasing the number of epochs from 12 to 20 and from 20 to 30 improves force MAE by 0.68 meV/Å and 0.36 meV/Å, respectively. As in Table 1(a), using the proposed separable S^2 activation and separable layer normalization improve force MAE by 1.09 meV/Å and 0.31 meV/Å. The improvement brought by the proposed architecture is comparable to that by training for longer.
>
> But as shown in Table 5 in the appendix, training one base model on S2EF-2M dataset takes 1412 GPU-hours, so it is expensive to repeat for all of Table 1(a) to compute error bars.
>
> > 2. [Weakness 2] Comparison between SO(3) convolutions with eSCN parameterization and tensor product parameterization.
>
> We provide the comparison in **General Response 8**.
>
> > 3. [Weakness 2] Comparison of EquiformerV1 and V2 when keeping other architectural hyperparameters fixed
>
> We provide the comparison between EquiformerV1 and EquiformerV2 using the OC20 S2EF 2M dataset in **General Response 7**.
>
> > 4. [Weakness 3] Include EquiformerV1 performance in Table 1.
>
> Thanks for the suggestion. Since all models use higher degrees (i.e., $L_{max}$ >= 4), which EquiformerV1 cannot use because of huge memory consumption, we will have another separate table summarizing the comparison.
>
> > 5. [Weakness 4] EquiformerV1 performance in Table 2 and 3.
>
> Since training EquiformerV2 on OC20 S2EF-All and S2EF-All+MD takes 20.5k and 37.7k GPU-hours and training EquiformerV1 can take similar amounts of time, we are unable to provide the results.
> Moreover, since increasing $L_{max}$ from 2 to 4 can significantly boost performances as shown in **General Response 7** and EquiformerV1 cannot use $L_{max}$ = 4, it can be less motivated to scale up a potentially less performant model.
>
> > 6. [Weakness 5] Differences in setups for IS2RE in Table 2 and EquiformerV1's Table 3?
>
> The results in Table 2 in this work are obtained by running relaxations. The results in Table 3 in Equiformer are obtained by directly predicting relaxed energies (direct approach). We compare the performances of EquiformerV1 and EquiformerV2 on IS2RE with direct approaches in **General Response 5**. We do not compare the IS2RE results with relaxation approaches since we already know that using $L_{max}$ = 4 can significantly improve force predictions and therefore IS2RE results and that EquiformerV1 cannot use $L_{max}$ = 4 as mentioned in **General Response 7**.
>
> > 7. [Weakness 6] Necessity of higher orders.
>
> We train different EquiformerV2 models of similar training time (~1440 GPU-hours) but with different $L_{max}$ and numbers of blocks on OC20 S2EF-2M dataset and summarize the comparison in Table VIII in the PDF. Increasing $L_{max}$ from 2 to 4 is clearly better than increasing depth. Increasing $L_{max}$ from 4 to 6 is on par with increasing depth.
>
> > 8. [Weakness 7] Benchmark EquiformerV2 on QM9.
>
> We provide additional results in **General Response 4**.
>
> > 9. [Question 1] The explanation of line 186, "the activation first converts vectors of all degrees to point samples on a sphere for each channel, applies unconstrained functions F to those samples, and finally convert them back to vectors."
>
> We provide the details in **Response 8 to Reviewer 6dKw**.
>
> > 10. [Question 2] EquiformerV2 results on OC20 IS2RE with direct setting.
>
> We provide the results in **General Response 5**.
>
> > 11. [Question 3] Justify "higher degrees can better capture angular resolution and directional information".
>
> We provide more details in **General Response 10**.
>
> > 12. [Question 4]  Was stochastic depth used in EquiformerV2?
>
> Yes. See Table 4 in appendix.
>
> > 13. [Question 5] Apply typical functions to rotated features during message passing.
>
> Typical functions refer to those we can use without considering equivariance or any constraint.
>
> > 14. [Question 6] Line 142: "eSCN convolutions go one step further and replace the remaining non-trivial paths of the SO(3) tensor product with an SO(2) linear operation to allow for additional parameters of interaction between +-m_0 without breaking equivariance."
>
> Line 142 corresponds to Line 436-439. eSCN convolutions directly use w^{\tilde} for learnable parameters. This allows removing the summation over degrees of filter (i.e., $L_{f}$) and removing Clebsch-Gordan coefficients. Since using w^{\tilde} is mathematically equivalent to using w as discussed in the work of eSCN, this will preserve equivariance.
>
> > 15. [Question 7] Line 175: why would f_{ij}^{(0)} be less well-normalized?
>
> We provide more details about attention re-normalization and separable layer normalization in **General Response 9**.
>
> > 16. [Question 8] Line 178: Overload in notation with a and a_{ij}.
>
> We will replace a with $w_{a}$ for better clarity.
>
> > 17. [Question 9] Line 210: No centering in the layer norm formulation.
>
> Adding centering to layer normalization for vectors of degrees > 0 can preserve equivariance. However, in Equiformer, using centering for degrees > 0 can hurt performance, and therefore, Equiformer does not include that. EquiformerV2 follows this practice.
>
> > 18. [Question 10] Line 210: small \epsilon in the new separable layer normalization?
>
> Yes.
>
> > 19. [Question 11] ​​Line 280: "Finally, these modifications enable training for longer without overfitting (row 7)" don't both models improve performance when trained for longer, not just the model in row 7?
>
> Yes, we will remove that for better clarity and just mention training for longer can keep improving the performance.
>
> > 20. [Question 12] Comparison of speed and FLOPs.
>
> Models are trained with the same training/inference pipeline. In **General Response 6**, we provide detailed comparisons of training time, training throughput, numbers of parameters between models in Table 2.

---

> > ### Comment · Reviewer_9yvD · 2023-08-14
> > **Thanks for the response**
> >
> > Thanks for the response -- I've increased my score to a 6.

---

> > > ### Author Response · Authors · 2023-08-14
> > >
> > > Thank you for increasing the score from 5 to 6.
> > >
> > > Please let us know if you have any other question.

---

### Official Review · Reviewer_GdTP · 2023-07-07

**Soundness:** 4 excellent
**Presentation:** 4 excellent
**Contribution:** 3 good
**Rating:** 5
**Confidence:** 5

**Summary:**

This paper provides a new equivariant graph neural networks named EquiformerV2 to enhance the original Equiformer performance. It uses four new modules. The first module is to use the convolution in the eSCN (https://arxiv.org/abs/2302.03655) to replace the depth-wise tensor production accelerating the speed. The second module is the separable $S^{2}$ activation using the spherical grids in SCN (https://arxiv.org/abs/2206.14331) to encourage the non-linearity. The third module is the separable layer normalization which uses the variance of all equivariant feature $\ell \geq 1$ to do normalization. The fourth module is the attention re-normalization using a layer normalization before the non-linear function of the attention score branch.

**Strengths:**

1. This paper is well organized and written. The figure clearly shows the modification on the model architecture.
2. The experiments in Table 2 supports the proposed model architecture can achieve SOTA performance on the OC20 All training sets as well as OC20 All+MD. For example, on OC20 All, test energy MAE is improved 13meV in S2ET test and 25meV in IS2RE test. Such improvement is great. These two datasets usually take extensive training time to perform experiments on them. From the Throughput metric, the EquiformerV2 has better training efficiency compared to current baselines.


**Weaknesses:**

1. As a suggestion, it will be better if efficiency study includes both training time and inference time. Computational complexity metric such as FLOPs or MACs can help measure the inference complexity.
2. The description of spherical grid is not very clear in the paper. Although it is introduced in the SCN paper, I think a brief introduction can help people understand why such operation can enhance the non-linearity.


**Questions:**

1. For the speed up of DFT calculations, could you briefly introduce the AdsorbML and explain why EquiformerV2 can be used to accelerate the DFT?
2. Is it possible to apply such efficient equivariant architecture in directly accelerating the DFT calculations such as the SchNorb (https://www.nature.com/articles/s41467-019-12875-2), PhiSNet (https://arxiv.org/abs/2106.02347), QHNet (https://arxiv.org/abs/2306.04922) and recent QH9 dataset (https://arxiv.org/abs/2306.09549)? Since the computational complexity is also a problem for these equivariant networks.
3. For the equivariance, since the equiformerV2 uses the spherical grids, I am curous about whether equiformerV2 is rigorous equivariant or approximate equivariant. If this architecture is approximate equivariant, I think it will be great if there are some experiments to verify the equivariance.

**Limitations:**

1. As an suggestion, to comprehensively study the improvement on original Equiformer, it will be better if there is a comprehensive experiments on the original datasets such as QM9.

---

> ### Author Rebuttal · Authors · 2023-08-10
>
> > 1. [Weakness 1] As a suggestion, it will be better if efficiency study includes both training time and inference time. Computational complexity metric such as FLOPs or MACs can help measure the inference complexity.
>
> We provide the comparison in **General Response 6**.
>
> > 2. [Weakness 2] The description of spherical grid is not very clear in the paper. Although it is introduced in the SCN paper, I think a brief introduction can help people understand why such an operation can enhance the non-linearity.
>
> We provide more details about S^2 activation in **Response 8 to Reviewer 6dKw**.
>
> Moreover, since the inner products between one channel of vectors of all degrees and the spherical harmonics projections of sampled points sum over all degrees, the conversion to 2D grid feature maps implicitly considers the information of all degrees. Therefore, S^2 activation, which converts equivariant features into 2D grid feature maps, uses the information of all degrees to determine the non-linearity. In contrast, gate activation only uses vectors of degree 0 to determine the non-linearity of vectors of higher degrees. More concretely, gate activation applies sigmoid to vectors of degree 0 to obtain non-linear weights and then multiply vectors of higher degrees with those non-linear weights. For tasks such as force predictions, where the information of degrees is critical, S^2 activation can be better than gate activation since S^2 activation uses all degrees to determine non-linearity.
>
> > 3. [Question 1] For the speed up of DFT calculations, could you briefly introduce the AdsorbML and explain why EquiformerV2 can be used to accelerate the DFT?
>
> We provide the details in **Response 5 to Reviewer 3XDb**.
>
> > 4. [Question 2] Is it possible to apply such efficient equivariant architecture in directly accelerating the DFT calculations such as the SchNorb, PhiSNet, QHNet and recent QH9 dataset? Since the computational complexity is also a problem for these equivariant networks.
>
> Yes, this is a great suggestion, and it is possible to apply EquiformerV2 to those problems, but implementing them would require substantial architectural changes and can be future work. Concretely, since they are predicting Hamiltonian matrices, they need complete graphs, where all nodes are connected by edges. Besides, in addition to node features, they need to maintain edge features beyond just for message passing. Finally, they also need inverse tensor products applied to edge features to predict Hamiltonian matrices at the output. These modifications are beyond the scope of this work, but we do agree that higher degrees with higher efficiency can be very helpful for predicting Hamiltonian matrices.
>
> > 5. [Question 3] For the equivariance, since the EquiformerV2 uses the spherical grids, I am curious about whether EquiformerV2 is rigorous equivariant or approximate equivariant. If this architecture is approximate equivariant, I think it will be great if there are some experiments to verify the equivariance.
>
> We provide the details in **Response 8 to Reviewer 6dKw**.
>
> eSCN empirically computes such errors in Figure 9 in their latest manuscript and shows that the errors of using $L_{max}$ = 6 and sampling resolution R = 18 are close to 0.2%, which is similar to the equivariance errors of tensor products in e3nn. The equivariance errors in e3nn are due to numerical precision. As long as we choose a high R, the equivariance errors can be empirically kept at the same level as the errors of those strictly equivariant operations.
>
> > 6. [Limitation] As a suggestion, to comprehensively study the improvement on original Equiformer, it will be better if there are comprehensive experiments on the original datasets such as QM9.
>
> We provide additional results on QM9 dataset in **General Response 4**, additional results on OC20 IS2RE dataset in **General Response 5**, and additional results on OC20 S2EF 2M dataset in **General Response 7**.
>
> In general, EquiformerV2 improves upon Equiformer when the targeted datasets or tasks such as OC20 S2EF and OC20 IS2RE with IS2RS auxiliary task require higher degrees or more expressivity. For the smaller datasets like QM9, the benefits of using stronger models are not very obvious. We note that this is consistent with the findings in previous works like Equiformer [1] (Section G in appendix) and Noisy Nodes [2].
>
> Additionally, we note that Row 1 in Table 1(a) does correspond to the architecture of the original Equiformer except that we use efficient SO(3) convolutions to incorporate higher degrees to isolate the effect of the proposed architectural improvements.
>
> [1] Liao et al. Equiformer: Equivariant Graph Attention Transformer for 3D Atomistic Graphs. ICLR 2023.
>
> [2] Godwin et al. Simple GNN Regularisation for 3D Molecular Property Prediction and Beyond. ICLR 2022.

---

### Official Review · Reviewer_6dKw · 2023-07-08

**Soundness:** 3 good
**Presentation:** 2 fair
**Contribution:** 3 good
**Rating:** 5
**Confidence:** 4

**Summary:**

In this paper, the authors proposed EquiformerV2, which is an equivariant network for 3D molecular modeling. The EquiformerV2 is built on the Equiformer with several architectural modifications: 1) replace SO(3) convolutions (tensor product operations) with efficient SO(2) counterparts from eSCN; 2) Attention Re-normalization; 3) Separable S2 activation; 4) separate Layer Normalization. These changes enable EquiformerV2 to achieve good performance on the large-scale OC20 benchmark and also the new AdsorbML dataset.

**Strengths:**

1. The targeted problem is of great interest to the community. The EquiformerV2 provides another attempt to enlarge the maximum degree of irreducible representations and obtain performance gains on large-scale DFT benchmarks.

2. Good empirical performance. On the OC20 benchmark, the EquiformerV2 achieves state-of-the-art performance on the Structure-to-Energy-Force task. The model trained on this task further serves as a force-field evaluator to achieve strong performance on IS2RS and IS2RE tasks. The EquiformerV2 outperforms the compared baselines on all these tasks, especially on force prediction. Additionally, it also improves the success rate a lot on the AdsorbML dataset.


**Weaknesses:**

1. The novelty of integrating the eSCN convolution and S^2 activation into the Equiformer is limited. Among the proposed architectural changes, the eSCN convolution is the key component to enable Equiformer to use irreducible representations of higher degrees, and the S^2 activation also replaces all non-linear activations. However, these design strategies should be mainly credited to the eSCN work.

2. The motivation for the other architectural modifications should be thoroughly clarified. First, lines 174-175 in Section 4.2 suggest the "less well-normalized" issue, which motivated the authors to propose re-normalization and Separable Layer Normalization. It is better to provide further quantitive evidence to reveal how such an issue affects the model's performance, and why these modifications could remove or mitigate such effect. Second, the authors proposed Separable S^2 Activation because the original S^2 activation would make the training process diverge. However, it is hard to understand why such separable modifications could make the training process stable. Is there any further essential reason behind such a phenomenon? It is suggested to provide further analysis on such modifications.

3. More analyses are necessary if the computational resources are acceptable. First, the authors did not provide a performance comparison between whether using the eSCN convolution. Although the eSCN convolution is equivalent to the SO(3) counterpart, the computation processes of these two parameterizations are different. Second, it is suggested to further report the number of model parameters and memory costs for the EquiformerV2 and the compared baselines in Table 2. Third, how does EquiformerV2 perform on the IS2RS and IS2RE using the direct setting (use or not the denoising setting) that is the same as EquiformerV1?

Overall, the major weakness of this work lies in the novelty, unclear motivations, and incomplete analyses. If the authors could well address the above concerns, I would like to increase my scores.



**Questions:**

1. Towards the claim "Higher degrees can better capture angular resolution and directional information, which is critical to accurate prediction of atomic energies and forces". [Lines 32-33, also see Lines 97-98], could you further provide explanations and evidence from deep learning molecular models?

2. How is the S^2 activation implemented? Is there any sampling process inducing randomness or incompleteness to make the module not strictly equivariant? If so, is there any measurement of such approximated equivariance?

3. Could you briefly introduce how the EquiformerV2 model accelerates the DFT calculation in AdsorbML? Are there further quantitative results on the error measure (e.g., MAE) between the EquiformerV2 predictions and DFT labels?

**Limitations:**

The authors carefully discuss the broader impact and limitations of this work.

---

> ### Author Rebuttal · Authors · 2023-08-10
>
> > 1. [Weakness 1] The novelty of integrating the eSCN convolution and S^2 activation into the Equiformer is limited.
>
> We would like to clarify that the contribution of this work is to investigate whether the design choices of previous equivariant Transformers, which consider only lower degrees, can scale well to higher degrees as mentioned in the section of abstract. Replacing original tensor products with eSCN convolutions is necessary to scale to higher degrees, and we are interested in what architectural changes we should have after using eSCN convolutions. As discussed in this work, we need to add one additional normalization in the attention blocks, modify activation functions and modify equivariant normalization layers to better leverage the benefits of higher degrees brought by eSCN convolutions. We also show that the original Equiformer architecture is sub-optimal when higher degrees are used as in Table 1(a).
>
> Additionally, we found that directly using S^2 activation does not result in stable training. We then proposed separable S^2 activation, which applies standard (typical) activation functions to invariant features (L = 0) and S^2 activation to equivariant features (L > 0).
>
> While these modifications might seem simple, we see simplicity as a strength, it takes extensive empirical investigation to attain it, and as noted by reviewer 9yvD, we hope that it doesn’t “detract from the significance of putting together the full model and showing that it improves performance on a benchmark.”.
>
> > 2. [Weakness 2] The motivation for attention re-normalization and separable layer normalization.
>
> We provide more details about attention re-normalization and separable layer normalization in **General Response 9**.
>
> > 3. [Weakness 2] Training instability of S^2 activation and motivation for separable S^2 activation.
>
> We found that row 3 (with S^2 activation) in Table 1(a) has 10X larger gradient norm than row 4 (with separable S^2 activation) after 25% of the training. The sudden increase in gradient norm results in training instability. In contrast, gate activation, where scalars are updated with typical SiLU activation and transformed on their own, does not have any training instability. Therefore, this motivates applying separate activation functions to scalars and vectors of degrees > 0. We will add this observation and motivation to the manuscript.
>
> > 4. [Weakness 3] Comparison between SO(3) convolutions with eSCN parameterization and tensor product parameterization.
>
> We provide the comparison in **General Response 8**.
>
> > 5. [Weakness 3] Second, it is suggested to further report the number of model parameters and memory costs for the EquiformerV2 and the compared baselines in Table 2.
>
> We provide the comparisons of training time, training throughput and numbers of parameters in **General Response 1**. We report the maximum batch size each model can use on a single V100 GPU (32GB) as below.
> | Model                       | Maximum batch size  |
> |-----------------------------|---------------------|
> | GemNet-OC-L                 | 6                   |
> | SCN L=6 K=16 (4-tap 2-band) | 4                   |
> | SCN L=8 K=20                | 8                   |
> | eSCN L = 6 K = 20           | 4                   |
> | EquiformerV2 (31M)          | 14                  |
> | EquiformerV2 (153M)         | 4                   |
>
> > 6. [Weakness 3] Comparison of EquiformerV1 and EquiformerV2 on OC20 IS2RE with a direct setting.
>
> We provide the results in **General Response 5**.
>
> > 7. [Question 1] Towards the claim "Higher degrees can better capture angular resolution and directional information, which is critical to accurate prediction of atomic energies and forces". [Lines 32-33, also see Lines 97-98], could you further provide explanations and evidence from deep learning molecular models?
>
> We provide more details about “higher degrees can better capture angular resolution and directional information” in **General Response 10**.
>
> > 8. [Question 2] Implementation of S^2 activation.
>
> Our implementation is the same as that in e3nn, SCN and eSCN.
> We uniformly sample a fixed set of points on a unit sphere along the dimensions of longitude (alpha) and latitude (beta).
> We set the resolutions R of alpha and beta to be 18 when $L_{max} = 6$, meaning that we will have 324 (=18 * 18) points.
> Once the points are sampled, they are kept the same during training and inference, and there is no randomness.
> For each point, we compute spherical harmonics projection of degrees up to $L_{max}$.
> We consider an equivariant feature of C channels and each channel contains vectors of all degrees from 0 to $L_{max}$.
> When performing S^2 activation, for each channel and for each sampled point, we compute the inner product between the vectors of all degrees contained in one channel of the equivariant feature and the spherical harmonics projections of a sampled point.
> This results in R * R * C values, where the first two dimensions (R * R) correspond to grid resolutions and the last corresponds to channels.
> They can be viewed as 2D grid feature maps and treated as scalars, and we can apply standard (typical) activation functions like SiLU or use standard linear layers performing feature aggregation along the channel dimension.
> After that, we project back to vectors of all degrees by multiplying those values with corresponding spherical harmonics projections of sampled points.
> Although there is sampling, Spherical CNNs and eSCN mention that as long as R is high, the equivariance error can be close to 0.
> eSCN empirically shows that the errors are similar to numerical errors of strictly equivariant operations.
>
> > 9. [Question 3] More details about AdsorbML and further quantitative results.
>
> Please refer to **Response 5 to Reviewer 3XDb** for details about AdsorbML.
> We provide more quantitative results **General Response 2**.

---

### Official Review · Reviewer_3XDb · 2023-07-16

**Soundness:** 3 good
**Presentation:** 3 good
**Contribution:** 3 good
**Rating:** 5
**Confidence:** 3

**Summary:**

This paper propose EquiformerV2, a advance verison based on Equiformer and eSCN structure extend to higher degree representations, which achieve better performance in force and energy tasks.

**Strengths:**

This paper is well-written and organized, presenting a clear and coherent structure throughout. The introduced EquiformerV2, an upgraded version of the original Equiformer, including three architectural improvements: attention re-normalization, separable S^2 activation, and separable layer normalization. These enhancements contribute to the SOTA performance in  OC20 dataset.
The proposed model achieves a high degree representation with efficiency, as outlined in the paper. The authors also provide a comprehensive ablation study to support the necessity of these modifications, effectively highlighting their respective contributions to the overall performance improvement.

**Weaknesses:**

* A few spelling errors. For instance, in Section 6, the word "acknolwdge" etc.  Along with any other mistakes found throughout the manuscript.

* The experiments conducted in this study primarily utilize the OC20 dataset. While this dataset is relevant, it is essential to note that there are various other DFT-based datasets available that could provide a more comprehensive evaluation of the proposed architecture. Such as OC22, OQMD[1,2], SPICE[3], and PCQM4Mv2 etc.

[1]  Saal, J. E., Kirklin, S., Aykol, M., Meredig, B., and Wolverton, C. "Materials Design and Discovery with High-Throughput Density Functional Theory: The Open Quantum Materials Database (OQMD)", JOM 65, 1501-1509 (2013).
[2]. Kirklin, S., Saal, J.E., Meredig, B., Thompson, A., Doak, J.W., Aykol, M., Rühl, S. and Wolverton, C. "The Open Quantum Materials Database (OQMD): assessing the accuracy of DFT formation energies", npj Computational Materials 1, 15010 (2015).
[3] SPICE, A Dataset of Drug-like Molecules and Peptides for Training Machine Learning Potentials


**Questions:**

* Would it be possible for you to include the results from the small dataset, as you mentioned? This would provide valuable insights into the performance and scalability of your proposed approach.
* More details about structural relaxations, specifically, the relax trajectories and time efficiency.

**Limitations:**

same above.

---

> ### Author Rebuttal · Authors · 2023-08-10
>
> > 1. [Weakness 1] A few spelling errors. For instance, in Section 6, the word "acknolwdge" etc. Along with any other mistakes found throughout the manuscript.
>
> Thanks for finding this. We will double check the paper and correct spelling mistakes if we find one.
>
> > 2. [Weakness 2] The experiments conducted in this study primarily utilize the OC20 dataset. While this dataset is relevant, it is essential to note that there are various other DFT-based datasets available that could provide a more comprehensive evaluation of the proposed architecture. Such as OC22, OQMD, SPICE, and PCQM4Mv2 etc.
>
> We provide additional results on OC22 dataset in **General Response 3**.
>
> > 3. [Question 1] Would it be possible for you to include the results from the small dataset, as you mentioned? This would provide valuable insights into the performance and scalability of your proposed approach.
>
> We provide additional results on QM9 dataset in **General Response 4** and additional results on IS2RE with a direct approach in **General Response 5**.
>
> > 4. [Question 2] More details about structural relaxations, specifically, the relax trajectories and time efficiency.
>
> Some relevant details of running relaxations can be found in Section C.2 in appendix. Specifically, after training models on OC20 S2EF dataset, we use LBFGS optimizer implemented in Open Catalyst GitHub repository to update the atomic positions given their atomwise forces predicted by trained models. Once the atomic positions are updated, we run models to predict the updated atomwise forces. The process of running relaxations, updating atomic positions and running trained models to predict atomwise forces, is repeated for a pre-defined number of steps or until the maximum predicted force per atom is less than 0.02 eV/Å.The predefined numbers of steps are 200 for OC20 IS2RE and IS2RS and 300 for AdsorbML. Additionally, the structures in the OC20 dataset have pre-defined sets for adsorbate atoms, surface atoms and subsurface atoms, and only the positions of first two types of atoms are updated during relaxations. The above setting is the same as previous works. Please refer to their paper [1] and GitHub repository [2] for more details and let us know if more specific details can be helpful. We additionally provide more details about AdsorbML [3] in the next response.
>
> Please let us know if there are any other details we can help provide on the relaxation trajectories.
>
> The amount of time required to run relaxations can also be found in Section C.2 in appendix. For the smaller version of EquiformerV2 trained on OC20 S2EF-All+MD dataset in **General Response 1**, the time for running relaxations is 240 GPU-hours for OC20 IS2RE and IS2RS and 298 GPU-hours for AdsorbML. The smaller version takes 4.48X and 3.61X less GPU-hours than the one reported in Table 2.
>
> Additionally, we compare GPU-seconds of different models in Table 3 required to run relaxations for AdsorbML in **General Response 2**. We note that the smaller version of EquiformerV2 in **General Response 1** is more accurate than other models and is 9.8X faster than SCN and 3.7X faster than GemNet-OC-MD-Large.
>
> [1] Chanussot et al. Open Catalyst 2020 (OC20) Dataset and Community Challenges. ACS Catalyst 2021.
>
> [2] https://github.com/Open-Catalyst-Project/ocp
>
> [3] Lan et al. AdsorbML: Accelerating Adsorption Energy Calculations with Machine Learning. ArXiv 2023.
>
> > 5. Additional details about AdsorbML.
>
> AdsorbML [1] aims at finding the global minimum of relaxed energy, or adsorption energy, given an adsorbate and a catalyst.
> The algorithm first generates some configurations of the adsorbate on the catalyst surface based on some heuristics or randomly, and it then runs relaxations for each of the resulting initial structures formed by those generated configurations.
> After running relaxations, some relaxed structures that do not satisfy predefined constraints (e.g., dissociations or desorptions) are removed, the relaxed energies of the remaining relaxed structures are calculated, and the lowest relaxed energy is considered the adsorption energy.
>
> Once we finish training models (e.g., EquiformerV2) on OC20 S2EF datasets, we can use the models for both relaxations and predictions of energies of relaxed structures.
> For relaxations, we use models to predict atomwise forces to update the atomic positions until the predicted forces are below a certain threshold or we reach a predefined number of optimization steps. Since originally the atomic positions are updated with compute-intensive DFT calculations, using ML models to replace DFT for force calculations can accelerate the process of AdsorbML.
> For predicting the energies of relaxed structures, we can either use single-point DFT (i.e., only one DFT calculation) or trained models.
> AdsorbML uses single-point DFT to predict the energies of relaxed structures by default since they find that it strikes a good balance between compute and accuracy.
>
> [1] Lan et al. AdsorbML: Accelerating Adsorption Energy Calculations with Machine Learning. ArXiv 2023.

---

> > ### Comment · Reviewer_3XDb · 2023-08-18
> >
> > thanks for your response, I decide to maintain my initial scores for your submission after full consideration.

---

> > > ### Author Response · Authors · 2023-08-18
> > >
> > > Thanks for your response. We believe we addressed all your comments, but please let us know if you have any other question.

---

### Author Rebuttal · Authors · 2023-08-10

Thank you for all the constructive feedback! We are glad reviewers found the writing clear (3XDb, GdTP, 9yvD) and the empirical results on OC20 and AdsorbML impressive (6dKw, GdTP, 9yvD, 3XDb). We address general questions here:

> 1. Smaller EquiformerV2

We trained a smaller version of EquiformerV2 on OC20 S2EF-All+MD dataset and include the results on OC20 and AdsorbML in the PDF (Tables I and II).

> 2. Energy MAE, speed-accuracy trade-off for AdsorbML.

See Table II in the PDF.

> 3. Results on the OC22 dataset

We train EquiformerV2 with $L_{max} = 6$, $M_{max} = 2$ and the number of blocks = 18 on OC22 S2EF-Total dataset and summarize the comparisons in Table III in the PDF.

> 4.  Results on the QM9 dataset.

We compare EquiformerV2 with $L_{max} = 4$, $M_{max} = 4$, number of blocks = 6 with EquiformerV1 in Table IV in the PDF. Unlike OC20 S2EF, using higher degrees ($L_{max}$ increased from 2 to 4) does not result in a significant improvement on QM9. This is not surprising as QM9 has fewer examples and smaller graphs, resulting in tasks that are simpler than the OC20 S2EF task and therefore not requiring higher degrees or better expressivity.

> 5. Results on OC20 IS2RE using a direct approach.

We compare EquiformerV1 and EquiformerV2 on OC20 IS2RE with and without IS2RS auxiliary task in Tables V in the PDF. Using higher degrees in EquiformerV2 does not improve performance on IS2RE. However, when using IS2RS, EquiformerV2 improves upon EquiformerV1. Higher degrees with better expressivity can lead to overfitting if the task (IS2RE only) does not require better expressivity [1]. For the task of IS2RE + IS2RS, where better expressivity does translate to better performance [1], EquiformerV2 clearly improves energy MAE on all splits.

[1] Godwin et al. ICLR 2022. https://arxiv.org/abs/2106.07971

> 6. Comparison of training time, inference speed, numbers of parameters and FLOPs between SCN, eSCN and EquiformerV2.

We compare training time, training throughput and number of parameters of different models and compare FLOPs between SCN, eSCN and EquiformerV2. We note that all the models contain node-wise and pair-wise representations without triplet representations or quadruplet representations as used in GemNet-OC. As the numbers of representations can depend on datasets, we consider the comparison between SCN, eSCN and EquiformerV2 for simplicity and fairness.
The comparison is in Table I in the PDF.

Table 5 in appendix already reported training time, inference speed, and number of parameters of EquiformerV2.

> 7. Comparison between EquiformerV1 and EquiformerV2 on OC20 S2EF 2M.

We train EquiformerV1 and EquiformerV2 with $L_{max} = 2$, the number of channels = 128 for each degree and the number of blocks = 8 on OC20 S2EF-2M dataset. The results are summarized in Table VI in the PDF. EquiformerV2 requires 2.3x less training time and achieves better force MAE but slightly worse energy MAE than EquiformerV1. For EquiformerV2, we additionally increase $L_{max}$ from 2 to 4 to show the further performance gain. We cannot train EquiformerV1 with $L_{max} > 2$ due to huge memory cost.

> 8. Comparison between SO(3) convolutions with eSCN parameterization and tensor product parameterization.

We use EquiformerV2 with $L_{max} = 6$, $M_{max} = 2$ and the number of blocks = 12 to compare SO(3) convolutions with different parameterizations. We summarize the results in Table VII in the PDF.

> 9. Clarification on attention re-normalization and separable layer normalization (LN).

We use $f_{ij}^{(0)}$ (line 170-173) as an example. $f_{ij}^{(0)}$ is obtained by performing a linear operation aggregating all C channels of m = 0 components from all degrees (0 to $L_{max}$). This can be viewed as a feature of C * ($L_{max} + 1$) channels. LN in Equiformer normalizes each degree independently, which is similar to first dividing channels into ($L_{max} + 1$) groups and then normalizing each group. This can ignore relative importance of different channels. For instance, if we have two groups, and channels of the first group are from a normal distribution and those of the second group from the same distribution but scaled up by a large magnitude, the second group can contain more information since its spread is larger. However, after the normalization, the two groups will collapse to the same distribution, and relative importance is not preserved. In contrast, if we use normalization considering all channels in all groups, relative importance is maintained. We posit that maintaining relative importance can make training more stable.

Besides, ViT-22B [1] also uses additional LN to queries and keys when performing dot product attention. The extra LN can prevent attention logits (inputs to softmax) from growing too large when scaling up channels and thus prevent one-hot attention weights (outputs of softmax). This applies to our case. When we increase $L_{max}$, we increase the number of channels $(C * (1 + L_{max}))$ of the feature, from which we obtain $f_{ij}^{(0)}$ with a linear operation. Therefore, if we apply the extra LN to $f_{ij}^{(0)}$, we can make sure the input to subsequent softmax does not grow too large when we increase $L_{max}$ and therefore prevent the undesired one-hot attention weights.

As in Table 1(a), we see clear improvements after adding attention re-normalization and separable LN.

We will add these to the manuscript.

[1] Dehghani et al. Scaling Vision Transformers to 22 Billion Parameters. ICML 2023.

> 10. More details about “higher degrees can better capture angular resolution and directional information”.

Like higher frequencies in a Fourier transform better approximate the underlying signal, higher angular frequencies or degrees of spherical harmonics offer higher fidelity for spherical functions. Several referenced works (e.g. NequIP, SCN, eSCN) show that higher degrees are helpful for energy and force predictions. Please let us know if more details are needed.

---

> ### Author Response · Authors · 2023-08-17
> **Updated Results of EquiformerV2 on QM9 Dataset and Overall Comparison to Equiformer(V1)**
>
> > 1. Updated results of EquiformerV2 on QM9 dataset.
>
> We provide the updated results of EquiformerV2 on all 12 tasks of the QM9 dataset below.
> EquiformerV2 achieves **better results** than Equiformer(V1) **on 9 out of the 12 tasks**, same result on 1 task and slightly worse on the other 2 tasks.
>
>
> |                | $\alpha$ | $\Delta \varepsilon$ | $\varepsilon_{HOMO}$ | $\varepsilon_{LUMO}$ | $\mu$    | $C_{\nu}$ | $U_0$    | $U$      | $H$      | $G$      | $R^2$    | $ZPVE$   |
> |----------------|----------|----------------------|----------------------|----------------------|----------|-----------|----------|----------|----------|----------|----------|----------|
> | Equiformer(V1) | **.046** | 30                   | 15                   | 14                   | .011     | **.023**  | 6.59     | 6.74     | 6.63     | 7.63     | .251     | **1.26** |
> | EquiformerV2   | .050     | **29**               | **14**               | **13**               | **.010** | **.023**  | **6.17** | **6.49** | **6.22** | **7.57** | **.186** | 1.47     |
>
>
> Additionally, we note that in the PDF in our **General Response**, we have complete comparisons between Equiformer(V1) and EquiformerV2 on (1) QM9 dataset, (2) OC20 S2EF-2M dataset, (3) OC20 IS2RE dataset without IS2RS auxiliary task, and (4) OC20 IS2RE dataset with IS2RS auxiliary task.
>
> Moreover, EquiformerV2 achieves **better results** than Equiformer(V1) on **(1) QM9 dataset**, **(2) OC20 S2EF-2M dataset**, and **(4) OC20 IS2RE dataset with IS2RS auxiliary task**.
>
> We would like to repeat again that the performance gain brought by higher degrees can depend on tasks and datasets.
> For larger datasets like OC20 S2EF-2M dataset, higher degrees clearly improves (Table VI and Table VIII).
> For smaller datasets like QM9 or OC20 IS2RE, higher degrees do not always bring significant improvement, but for these cases, we can pre-train on larger datasets or incorporate auxiliary tasks [1] or denoising objectives [2] to leverage the better expressivity (see the section of limitations (Line 347-Line 352)).
> Indeed, we show that for OC20 IS2RE dataset, using higher degrees results in overfitting, but when we use IS2RS auxiliary task, EquiformerV2 clearly improves upon Equiformer(V1) as in Table V.
>
>
> [1] Godwin et al. Simple GNN Regularisation for 3D Molecular Property Prediction and Beyond. ICLR 2022.
>
> [2] Zaidi et al. Pre-training via Denoising for Molecular Property Prediction. ICLR 2023.

---

### Decision · Program_Chairs · 2023-09-21

**Decision:**

Reject

**Comment:**

This paper proposed EquiformerV2, which is built on the Equiformer with several architectural modifications, including an efficient convolution from eSCN; 2) Attention Re-normalization; 3) Separable S2 activation from SCN; 4) separate Layer Normalization. These changes enable EquiformerV2 to achieve good performance on the large-scale OC20 benchmark and also the new AdsorbML dataset.

The authors provided comprehensive empirical analysis on a wide range of datasets during rebuttal. However, as acknowledged by reviewers 6dkw and gdtp, this work is more like an experimental report of building a combination of existing modules, which lacks enough technical insight and a deep understanding of molecular modeling. Given the competitive submissions in the pool, I recommend rejection and hope the authors can address the issues in the future.